# Antidiabetic Effects of Bisamide Derivative of Dicarboxylic Acid in Metabolic Disorders

**DOI:** 10.3390/ijms21030991

**Published:** 2020-02-03

**Authors:** Angelina Vladimirovna Pakhomova, Vladimir Evgenievich Nebolsin, Olga Victorovna Pershina, Vyacheslav Andreevich Krupin, Lubov Alexandrovna Sandrikina, Edgar Sergeevich Pan, Natalia Nicolaevna Ermakova, Olga Evgenevna Vaizova, Darius Widera, Wolf-Dieter Grimm, Viacheslav Yur’evich Kravtsov, Sergey Alexandrovich Afanasiev, Sergey Georgievich Morozov, Aslan Amirkhanovich Kubatiev, Alexander Mikhaylovich Dygai, Evgenii Germanovich Skurikhin

**Affiliations:** 1Laboratory of Regenerative Pharmacology, Goldberg ED Research Institute of Pharmacology and Regenerative Medicine, Tomsk National Research Medical Centre of the Russian Academy of Sciences, Lenin, 3, 634028 Tomsk, Russiaovpershina@gmail.com (O.V.P.); vakrupin88@gmail.com (V.A.K.); ermolaeva_la@mail.ru (L.A.S.); artifexpan@gmail.com (E.S.P.); nejela@mail.ru (N.N.E.); amdygay@gmail.com (A.M.D.); 2Obschestvo s ogranichennoi otvetstvennostiyu “PHARMENTERPRISES”, 143026 Moscow, Russia; nve1970@mail.ru; 3Department of Pharmacology, Siberian State Medical University, 634050 Tomsk, Russia; vaizova@mail.ru; 4Stem Cell Biology and Regenerative Medicine Group, School of Pharmacy, University of Reading, Whiteknights campus, Reading RG6 6UB, UK; d.widera@reading.ac.uk; 5Periodontology, Department of Dental Medicine, Faculty of Health, University of Witten/Herdecke, Alfred-Herrhausen-Straße 50, 58 448 Witten, Germany; prof_wolf.grimm@yahoo.de; 6Biology Department, Military Medical Academy, 199004 Saint-Petersburg, Russia; kvyspb@mail.ru; 7Cardiology Research Institute, Tomsk National Research Medical Centre, Russian Academy of Sciences, 634012 Tomsk, Russia; tursky@cardio-tomsk.ru; 8Institute of General Pathology and Pathophysiology, 125315 Moscow, Russia; biopharm@list.ru (S.G.M.); niiopp@mail.ru (A.A.K.)

**Keywords:** metabolic disorders, diabetes, tissue-specific stem cells, Bisamide Derivative of Dicarboxylic Acid, regeneration

## Abstract

In clinical practice, the metabolic syndrome can lead to multiple complications, including diabetes. It remains unclear which component of the metabolic syndrome (obesity, inflammation, hyperglycemia, or insulin resistance) has the strongest inhibitory effect on stem cells involved in beta cell regeneration. This makes it challenging to develop effective treatment options for complications such as diabetes. In our study, experiments were performed on male C57BL/6 mice where metabolic disorders have been introduced experimentally by a combination of streptozotocin-treatment and a high-fat diet. We evaluated the biological effects of Bisamide Derivative of Dicarboxylic Acid (BDDA) and its impact on pancreatic stem cells in vivo. To assess the impact of BDDA, we applied a combination of histological and biochemical methods along with a cytometric analysis of stem cell and progenitor cell markers. We show that in mice with metabolic disorders, BDDA has a positive effect on lipid and glucose metabolism. The pancreatic restoration was associated with a decrease of the inhibitory effects of inflammation and obesity factors on pancreatic stem cells. Our data shows that BDDA increases the number of pancreatic stem cells. Thus, BDDA could be used as a new compound for treating complication of the metabolic syndrome such as diabetes.

## 1. Introduction

By definition of the International Association of Diabetes, metabolic syndrome (MS) is a combination of abdominal obesity, insulin resistance, hyperglycemia, dyslipidemia, arterial hypertension, disruption the hemostatic system, and chronic subclinical inflammation [1,2]. MS is a widespread complex of symptoms which is pandemic in industrialized countries (e.g., in the USA, Germany, France, New Zealand, and Australia) where it is found in every fifth adult person. The prevalence of MS is expected to increase by 50% in the next 25 years [3,4]. A major concern is a steady increase in the frequency of MS among adolescents and young people [5]. The main etiological factors of MS are genetic predisposition, excessive fat intake, and hypodynamia [6].

The clinical significance of MS lies in its complications, especially in combination with other diseases where it accelerates their development and progression [7,8]. In this context, MS can lead to serious complications such as cardiovascular disease, type 2 diabetes, osteoarthritis, sleep apnea, non-alcoholic steatohepatitis, respiratory problems, malignant tumors, and type 2 diabetes [9,10,11,12,13]. There is a linear relationship between the body mass index (BMI) and mortality from coronary heart disease (CHD), stroke, and diabetes [14,15].

Currently, there is little progress in developing curative therapies for complications of MS. A possible alternative to hormonal therapy of diabetic complications in patients with MS could be a selective drug affecting endogenous tissue-specific stem and progenitor cells. In such scenario, in addition the pharmacologically active molecule, a specific cellular target needs to be defined. Based on the findings from our previous studies suggesting that Bisamide Derivative of Dicarboxylic Acid (BDDA) has regenerative properties [16], we aimed to assess the efficacy of BDDA in metabolic disorders and their complications such as diabetes in a mouse model of metabolic disorders (MD). In addition, we evaluated the effects of BDDA on pancreatic stem and progenitor cells as potential cellular targets of BDDA. 

## 2. Results and Discussion

### 2.1. The Effect of BDDA on Serum Lipid Profile in MD

The disturbances in fat metabolism were accompanied by hyperglycemia, abnormal (glucose tolerance test) GTT, elevated serum Gastric inhibitory polypeptide (GIP), and insulin. Overall, these changes correspond to the clinical pattern of metabolic disorders (MD). Streptozotocin and high-fat diet caused an increase in the concentration of triglycerides (TG) (compared with intact values of 1.38 ± 0.15 by 15%) and very low-density lipoproteins (VLDL) (compared with intact values 0.41 ± 0.08 by 38%) in the serum of male C57BL/6 mice (group 2—mice with metabolic disorders) compared to the intact control (group 1) on d70, with an increase in the atherogenic index (by 43%) (Figure 1). In contrast, the level of high-density lipoproteins (HDL) significantly decreased by 15% (compared with intact values 2.41 ± 0.09). The BDDA introduction reduced the concentration of TG (by 8%), LDL (29%) and VLDL (by 16%) in serum, the atherogenic index (by 25%) in mice of group 3 (mice with MD treated with BDDA) on d70 compared with group 2 mice. At the same time, we observed an increase in the level of serum HDL (by 6.5%). In addition, we studied the ratio of triglycerides to high-density lipoproteins (TG/HDL) [17,18]. As evident in Figure 1E, the parameter was significantly higher in mice with MD compared to the control group (1.5 fold). The BDDA treatment reduced the ratio of TG/HDL in group with MD compared to the controls.

### 2.2. The Effect of BDDA on Blood Glucose, GTT, and Insulin in Blood Serum

Simultaneous with fat metabolism disorder in mice of group 2, hyperglycemia was registered (the d42, d49, d56, d63, and d70) and a failure of the glucose tolerance test (d70) (Figure 2A,B). Serum insulin concentrations increased significantly (d70) (Figure 2C). On d70, the Caro index and Homeostasis Model Assessment of Insulin Resistance (HOMA-IR) were assessed (Figure 3) [19]. Tissue insulin sensitivity was determined by Quantitative Insulin Sensitivity Check Index (QUICKI) (Figure 3C) [20]. Data presented in Figure 3 that shows that tissue sensitivity to glucose changed, and insulin resistance developed in group 2. Introduction of BDDA reduced the levels of glucose in blood of mice on d56, d63, d70 and did not affect GTT in MD (Figure 2B), while it significantly reduced the level of serum insulin on d70 (Figure 2C).

### 2.3. The Effect of BDDA on Serum Cytokine Profile 

As anticipated, single administration of streptozotocin and long-term fat diet caused significant changes in the cytokine levels. Figure 4 shows that in group 2, the concentration of IL-4, IL-17, interferon-gamma (IFN-gamma), erythropoietin (EPO), and GIP in serum increased compared to group 1. In contrast, the levels of IL-1β, IL-1ra, IL-5, IL-23, and tumor necrosis factor-α (TNF-α) decreased (Figure 4). After introduction of BDDA, we observed increased serum levels of IL-1ra, IL-4, IL-5, IL-13, IL-23, and GLP-1 in mice within group 3 compared to animals in group 2. Moreover, the concentration of IL-17, IFN-gamma, and EPO was reduced (Figure 4). 

### 2.4. The Effect of BDDA on Tissue Morphology

Hematoxylin and eosin staining revealed pathological changes in the pancreas of mice with metabolic disorders on d49 and d70. Edema and hyperemia of the exocrine part of the pancreas were detected in addition to a small and medium-drop fat dystrophy of acinar cells, thickening and growth of interlobular septa, and an infiltration of islet tissue by inflammatory cells (Figure 5A,B). While the number of islets of Langerhans (53%) and the number of islet cells (53%) decreased in the group, the area of the islets of Langerhans was reduced by 52% compared group 1. In addition, the number of pyknotic cells increased 2.8 fold (Figure 5D). Notably, the pathological changes on d70 were more pronounced than on d49.

### 2.5. Flow Cytometric Analysis of Cells

On d70, multipotent beta cell progenitors (CD45^–^TER119^–^c-kit^–^Flk-1^–^), oligopotent beta cells precursors (CD45^–^TER119^–^CD133^+^CD49f^low^), and maturing PDX1^+^ beta cells in the pancreas were studied. Metabolic disorders reduced the number of PDX1^+^ beta cells and multipotent beta cell progenitors in the pancreas of group 2 compared to group 1, while the number of oligopotent beta cell precursors increased significantly (Figure 6). Application of BDDA increased the numbers of multipotent and oligopotent beta cell precursors and PDX1^+^- beta cells in the pancreas of group 3 mice compared with group 1 (Figure 6).

### 2.6. Discussion

Development of MS is associated with obesity, inflammation, and impaired tissue susceptibility to insulin action (insulin resistance). All these factors play a significant role in the development of cardiovascular diseases (arterial hypertension, coronary heart disease) and atherosclerosis. The effect of MS on many other organs becomes more and more obvious with the growth of MS problems in modern society. To date, in clinical practice, complications of MS such as type 2 diabetes are increasingly noted [21,22,23]. Several independent research groups have shown the emergence of insulin resistance in mice converted to a high-fat diet [24,25,26]. In the present study, we have shown that mice who received a fat diet have excessive deposition of visceral fat (data not are shown), an increased body mass index (BMI) (Appendix A in the Supplementary Material). Moreover, their lipid spectrum is disturbed: the level of "bad" cholesterol (LDL, VLDL) increases and the level of "good" cholesterol (HDL) decreases. However, there is no significant shift in the cytokine profile towards inflammatory mediators (Figure 4). On the contrary, in this model, we observed a decrease in the levels of inflammatory mediators in the serum (with the exception of IL-17 and IFN-gamma), and an increase in the levels of erythropoietin and GIP (Figure 4).

A high concentration of free fatty acids leads to the increased production of glucose by the liver (gluconeogenesis) and disruption of glucose transport inside the cells [27]. According to our data, animals with dyslipidemia develop hyperglycemia and insulin resistance on d49 (Figure 1 and Figure 2). We observed a consolidation of MD in male mice on high-fat diet on d70. A degeneration of Langerhans islands, up to necrotic changes, and a decrease in the number of islet PDX1^+^ beta cells were observed in group 2 (Figure 6). Chronic hyperglycemia and dyslipidemia is known to mediate excess production of reactive oxygen species and to an activation of caspases inhibiting insulin secretion, which eventually leads to apoptosis of pancreatic beta cells [28]. 

Pancreatic stem cells are involved in the neogenesis of producing and secreting beta cells. Studying the potential recovery of the population of beta cells in MD, we noted a significant (3 -fold) increases in oligopotent beta cell precursors (CD45^–^TER119^–^CD133^+^CD49f^low^) in the pancreas of mice with MD (group 2) (Figure 6). In an earlier study, we showed differentiation of oligopotent precursors into insulin-producing and insulin-secreting beta cells in type 1 diabetes [29]. Thus, we hypothesize that pancreatic stem cells are involved in the process of regeneration of beta cells in MD.

Several studies reported that MS and its components such as obesity, insulin resistance and dyslipidemia are associated with systemic inflammation, a process of which is preceded by oxidative stress with lipid peroxidation [30,31,32,33] and abnormal changes in the cytokine profile [34,35,36]. In contrast, other studies did not find changes in blood levels of TNF-α, IL-6, and IL-10 and no differential expression of the TNF-α in adipose tissue, liver, or skeletal muscle of obese animals [24,25,26]. Thus, we suggest that characterization of MD should consider the ratio of mediators of inflammation and anti-inflammation and take both serum and tissue cytokines into account.

Numerous complications of MS including diabetes are difficult to treat using conventional monotherapies. Treatment of patients requires a wide range of drugs and, usual, their application does not lead to the desired outcome. In our study, application of BDDA improved serum lipid metabolism and reduced the mass of visceral fat (data not shown), caused a hypoglycemic effect and had a positive effect on the insulin metabolism (Figure 1 and Figure 2). In this respect, we observed a restoration of the islet compartment. Fat dystrophy of acinar cells was less pronounced than in untreated mice with MD (Figure 5). This positive impact of BDDA on MD and its complications in male C57BL/6 mice was unexpected. We hypothesized that the effects of BDDA are mediated by tissue-specific stem cells. Here, according to our flow cytometry data, the number of pancreatic stem cells and maturing PDX1+ beta cells increases dramatically in group 3 mice (Figure 6).

Taken together, we present evidence for the efficacy of BDDA in treating the symptoms of obesity and diabetes, as well as stimulating the regeneration of the damaged pancreas. The positive effect of the treatment could be partially explained by a disruption of the causal relationship between obesity and inflammation, as well as obesity, insulin resistance, and diabetes with an increase in proliferation and differentiation of oligopotent beta cell precursors into insulin-producing beta cells as a cellular mechanism.

## 3. Materials and Methods

### 3.1. Animals

The experiments were performed on male C57BL/6 mice obtained from the nursery of the Surgical Bio-modeling Department of the Goldberg ED Research Institute of Pharmacology and Regenerative Medicine (veterinary certificate available), housed under pathogen-free conditions with food and water ad libitum. All animal experiments were carried out in accordance with the European Convention on the protection of vertebrates used in experiments or for other scientific purposes. The study was approved by the laboratory animal control Committee of the Goldberg ED Research Institute of Pharmacology and Regenerative Medicine, Tomsk NRMC (IACUC Protocol No. 114062016). 

### 3.2. Modeling of Metabolic Disorders

Experimental MD were modeled by a single subcutaneous injection of streptozotocin into the withers (Sigma, St. Louis, MO, USA) at a dose of 200 mg/kg in 30 µl of phosphate buffer one day after birth and the special diet administered on postnatal days 28–70, which was enriched with heavy saturated fats (Ssniff EF R / M with 30% Fat, Soest, Germany, ref. No. E15116-34) [37]. The introduction of streptozotocin was taken for the d0. Mice were co-housed (5–6 mice per cage) and entrained to a reverse 12 h light: 12 h dark cycle. During the 28 days, mice had ad libitum access to an unpurified standard rodent chow. The 20 mice were maintained on the standard chow diet for the entire study and served as a healthy control group. The remaining mice were given chow standard diet ad libitum within 28 days then they had fed a high-fat diet for six weeks to induce metabolic disorders (Figure 7).

### 3.3. Pharmacological Compound

The pharmacological compound Bisamide Derivative of Dicarboxylic Acid (BDDA), chemical formular N1,N5-bis [2-(1H-imidazole-2-Il)ethyl]glutaramide, was provided by the General Director of the "PHARMENTERPRISES" Ltd Mr. Nebolsin V. E. (Figure 8) [16].

BDDA was dissolved in sterile distilled water. BDDA was administered intragastrically daily at a dose of 10 mg/kg on d49—d70. Control animals received the solvent in the equivalent volume.

### 3.4. Experimental Groups

Mice without metabolic disorders receiving saline solution formed a control group (group 1 – control) (Table 1). Mice with metabolic disorders were divided into two experimental groups: mice with metabolic disorders (group 2—metabolic disorder) and mice with metabolic disorders treated with BDDA (group 3—metabolic disorders treated with BDDA). All the mice were sacrificed with CO_2_ on d70 after streptozotocin introduction.

### 3.5. Blood Glucose Test, Intraperitoneal Glucose Tolerance Test (GTT)

Blood glucose levels were measured using a glucometer (Accu-Chek Performa Nano (Roche Diagnostics GmbH, Mannheim, Germany)) on the d28, d35, d42, d49, d56, d63, and d70. 

On d70 a glucose tolerance test was performed. Measurement of the initial level of glucose in the blood of animals was performed after 12 hours of food deprivation. After that, glucose was administered intraperitoneally (D-glucose, Sigma, St. Louis, MO, USA) at a dose of 2 g/kg. Blood samples to study glucose level were taken 30, 60, 90, 120, 150 min after glucose administration [38].

### 3.6. Parameters Connected with Glucose Homeostasis in the Serum

Glucose level in the serum was evaluated with a glucometer Accu-Chek Performa Nano (Roche Diagnostics GmbH, Mannheim, Germany). Insulin levels in the serum were measured with the ELISA kit (Cusabio Biotech CO., LTD, China). On d70 the Caro index and Homeostasis Model Assessment of Insulin Resistance (HOMA-IR) were estimated using the following formulas [19]: Caro = GN/IN and HOMA−IR= (IN × IG)/22.5,
where IN-insulin is fasting, IU/ml; GN—fasting glucose, mmol/L.

Tissue insulin sensitivity was determined by Quantitative Insulin Sensitivity Check Index (QUICKI) [20]:QUICKI = 1/(login + loggn)

### 3.7. Lipid Profile Determination

The lipid profile was determined on the d70. Blood samples were taken from each animal in tubes without additives, kept at room temperature for 30 minutes and then centrifuged at 300g for 30 minutes. Serum was separated and used to study lipid profile parameters. The concentration of cholesterol and triglycerides (TG) was determined by direct enzymatic methods using BioSystems reagents (Barcelona, Spain) in accordance with the instructions for use. Fractions of high-density lipoproteins (HDL), low-density lipoproteins (LDL) and very low-density lipoproteins (VLDL) were precipitated with phosphovolframate and polyvinyl sulfate respectively, and then their concentration was determined by the level of residual cholesterol. All the results were expressed in mmol/L. Additionally, atherogenicity index (IA) or atherogenicity coefficient (KA) were determined. TG/HDL was assessed, as previously described in [17,18].

### 3.8. Enzyme-Linked Immunosorbent Assay

The concentration of insulin, IL-1ra, IL-1beta, IL- 4, 5, 13, 17, and 23, TNF-alpha, IFN-gamma, EPO, GIP, and GLP-1 in the serum was determined by the enzyme-linked immunosorbent assay (ELISA) in accordance with the instructions for use (Cusabio Biotech CO., LTD, China).

### 3.9. Morphological Assessment of the Pancreas

The morphological assessment of the pancreas was carried out on the d70. For this purpose, the tissues were fixed in a 10% solution of neutral formalin, processed through ascending concentrations alcohols to xylene and poured into paraffin. Dewaxed, 5 µm thick sections were stained with hematoxylin and eosin [39,40]. Micro-preparations from each animal were examined under a light microscope Axio Lab.A1 (Carl Zeiss, MicroImaging GmbH; Göttingen, Germany) at 100× and 400× magnifications.

Hematoxylin-eosin was performed to assess pancreas architecture and potential presence of inflammatory cells (neutrophils, lymphocytes, and macrophages). The area of the islets of Langerhans and the total number of cells and the number of pyknotic cells in the islets were determined on the pancreas preparations [41,42]. 

In addition, the presence of inflammatory infiltrate and the state of the microvasculature were assessed on preparations of the pancreas.

### 3.10. Flow cytometric Analysis

Mononuclears from pancreas were obtained as previously described methods on d70 [37]. Expression of receptors on the surface membranes of mice mononuclear cells was analyzed using surface markers. Briefly, cell suspensions were stained with fluorophore-conjugated monoclonal antibodies: CD45, CD49f, CD117 (c-kit), CD309 (Flk-1), TER119, and PDX1 (all Becton Dickinson, San Jose, CA, USA), and CD133 (Miltenyi Biotec GmbH, Germany). Appropriate isotype controls were used. Labeled cells were thoroughly washed with phosphate buffered saline (PBS) and analyzed on FACSCanto II (Becton Dickinson, San Jose, CA, USA) using FACS Diva software. At least 100,000 events were recorded for each condition.

### 3.11. Statistical Analysis

Statistical analysis was performed using SPSS statistical software (version 15.0, SPSS Inc., Chicago, IL, USA). Data were analyzed and presented as means ± standard error of mean. A two-sided unpaired Student t-test (for parametric data) or Mann-Whitney test (for nonparametric data) was used according to distribution. A P value of less than 0.05 (by two-tailed testing) was considered an indicator of statistical significance.

## 4. Conclusions

Overall, we show that BDDA improves the physiological functions of the pancreas from animals with metabolic disorders. This suggests that BDDA might be a promising lead compound to create a drug to stimulate tissue regeneration and the recovery of diminished tissue and organ function in patients with metabolic disorders.

## Figures and Tables

**Figure 1 ijms-21-00991-f001:**
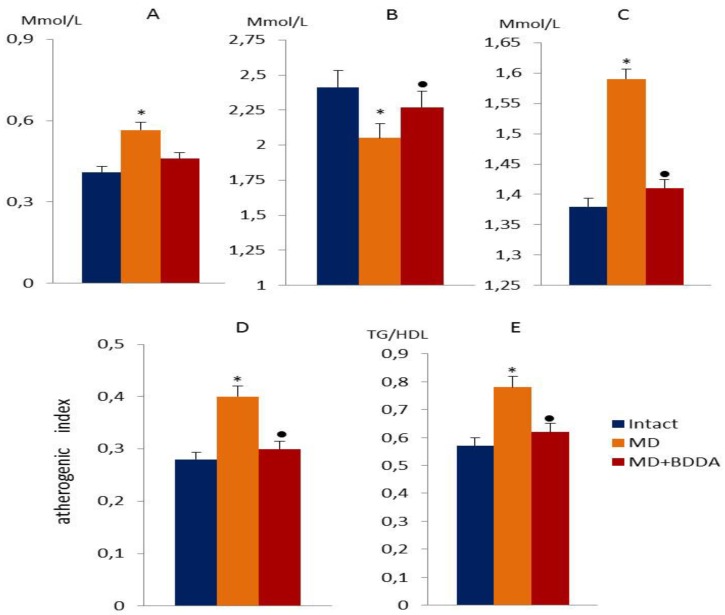
Lipid profile measurements in the blood of male C57BL/6 mice at d70. (**A**) The very low-density lipoprotein level (Mmol/L); (**B**) high-density lipoprotein level (Mmol/L); (**C**) the level of triglycerides in serum (Mmol/L); (**D**) the atherogenic index; (**E**) the ratio of triglycerides to high-density lipoproteins (TG/HDL). Groups: Intact—a control group from intact mice, MD—mice with MD, MD + BDDA—mice with MD treated BDDA. *: significance of difference compared with intact (*p* < 0.05); ●: significance of difference compared with the MD group (*p* < 0.05).

**Figure 2 ijms-21-00991-f002:**
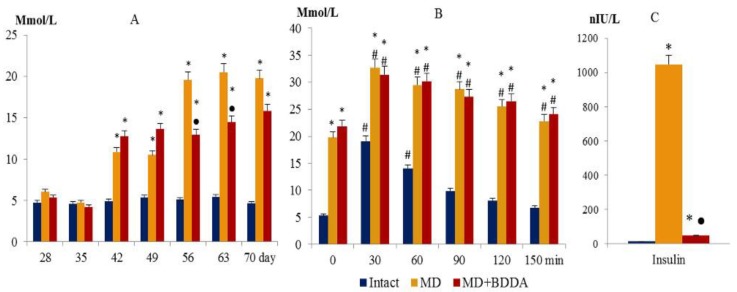
Blood glucose level (**A**), glucose tolerance test (**B**), and insulin level in serum of mice at d70 (**C**). Groups: intact—a control group from intact mice, MD—mice with MD, MD + BDDA—mice with MD treated BDDA. Results are presented as the mean ± SEM. *: significance of difference compared with intact (*p* < 0.05); ●: significance of difference compared with the MD group (p < 0.05). #: significance of difference compared with the baseline (0) (*p* < 0.05).

**Figure 3 ijms-21-00991-f003:**
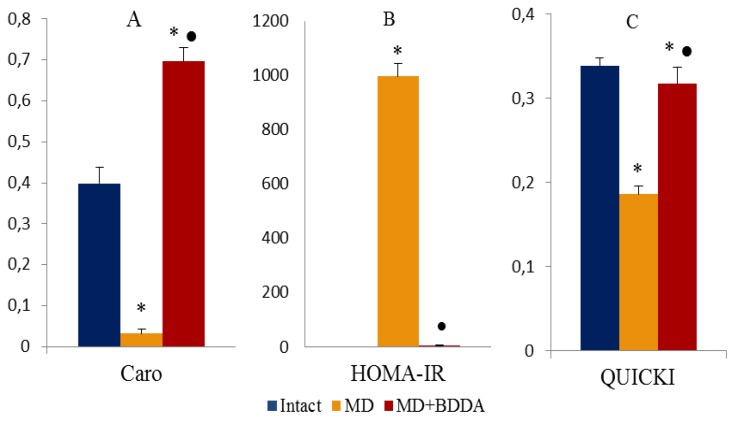
Caro (**A**) indices and HOMA-IR (**B**), QUICKI (**C**), UE were measured at d70. The Caro index and Homeostasis Model Assessment of Insulin Resistance (HOMA-IR) were estimated using the following formulas: Caro = GN/IN and HOMA−IR= (IN × IG)/22.5, where IN-insulin is fasting, IU/ml; GN—fasting glucose, Mmol/L. Groups: intact—a control group from intact mice, MD—mice with MD, MD+BDDA—mice with MD treated BDDA. *: significance of difference compared with controls (*p* < 0.05); ●: significance of difference compared with the MD group (*p* < 0.05).

**Figure 4 ijms-21-00991-f004:**
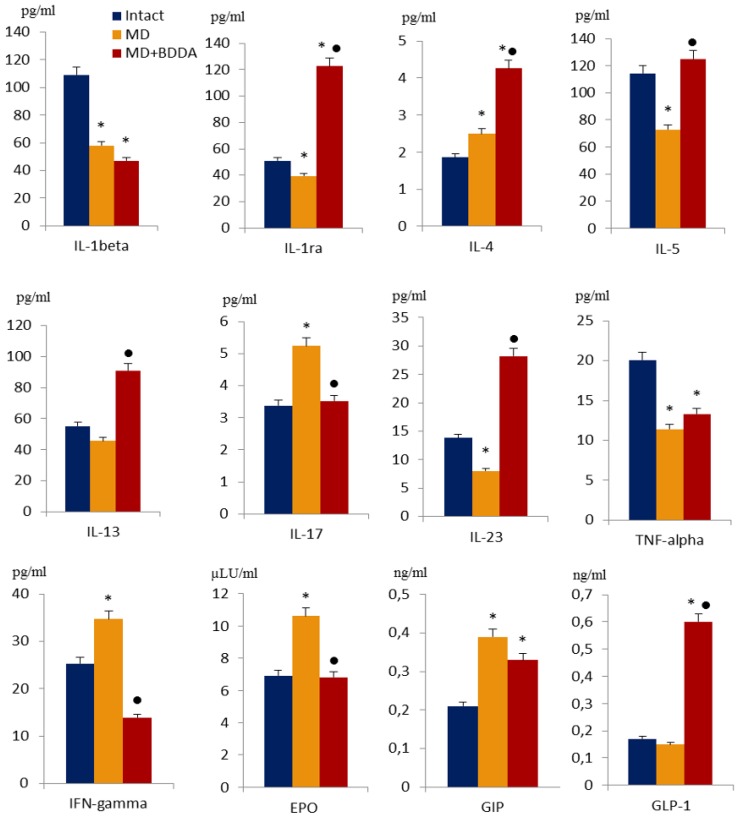
The level of interleukins (1beta, 4, 5, 13, 17, 23), IL-1ra, TNF-alpha, IFN-gamma, EPO, GIP and GLP-1 in the serum of male C57BL/6 mice at d70. Groups: intact—a control group from intact mice, MD—mice with metabolic disorders (MD), MD + BDDA—mice with MD treated BDDA. Results are presented as the mean ± SEM. *: significance of difference compared with intact (p<0.05); ●: significance of difference compared with the MD group (*p* < 0.05).

**Figure 5 ijms-21-00991-f005:**
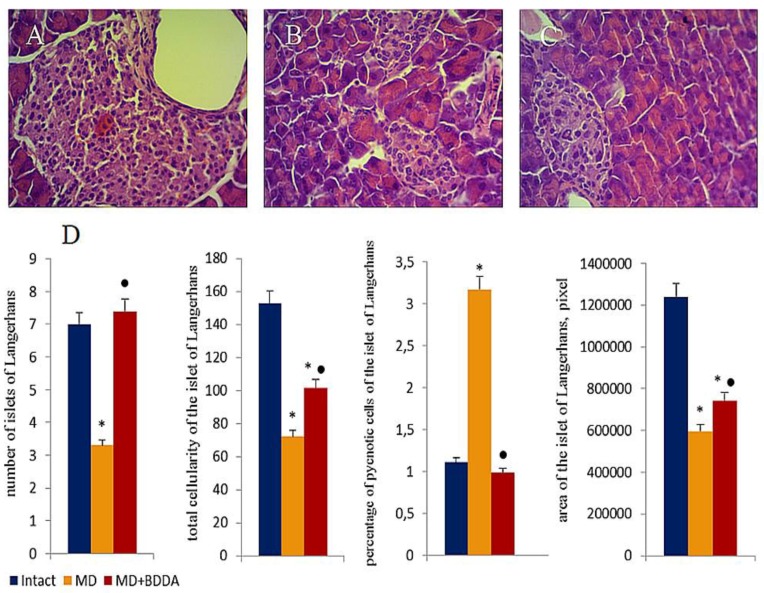
Photomicrographs of representative pancreas sections obtained from male C57BL/6 mice at d70. Tissues were stained with haematoxylin-eosin. (**A**) Sections from control group; (**B**) Sections from mice with metabolic disorders (MD); (**C**) Sections from mice with MD treated with BDDA from d49-d70. Langerhans islets were examined morphologically by measuring the islet area, total islet cells and the amount of pyknotic cells. At least 10 photomicrographs of the pancreas tissue at × 400 magnification were taken for each experimental animal from all experimental groups. (**D**) Number of Langerhans islets, total cellularity of the Langerhans islets, percentage of pyknotic cells in Langerhans islets and the area Langerhans islets. Groups: intact—a control group from intact mice, MD—mice with metabolic disorders (MD), MD+BDDA—mice with MD treated by BDDA. Results are presented as the mean ± SEM. *: significance of difference compared with control (*p* < 0.05); ●: significance of difference compared with the MD group (*p* < 0.05).

**Figure 6 ijms-21-00991-f006:**
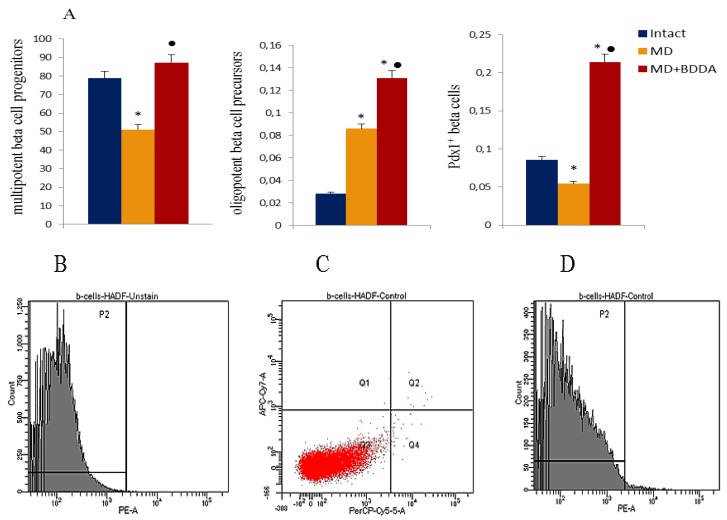
Characterization of multipotent beta cell progenitors (CD45^-^TER119^-^c-kit^-^Flk-1^-^), oligopotent beta cell precursors (CD45^-^TER119^-^CD133^+^CD49f^low^) and PDX1^+^ beta cells isolated from pancreata of male C57BL/6 mice on d70 (**A**). Groups: intact—a control group from intact mice, MD—mice with MD, MD + BDDA mice with MD treated BDDA. Results are presented as mean ± SEM. *: significance of difference compared with control (*p* < 0.05); ●: significance of difference compared with the MS group (*p* < 0.05). Cells were analyzed by flow cytometry using antibodies against mouse CD45, TER119, CD133, CD49f, c-kit, Flk-1 and PDX1. The population of CD45 and TER119 negative cells was defined as c-kit and Flk-1 negative and positive for CD133 and CD49f at the same time in the two specimens (specimen 1 - subpopulation of cells with phenotype CD45^-^TER119^-^c-kit^-^Flk-1^-^ and specimen 2 - subpopulation of cells with phenotype CD45^-^TER119^-^CD133^+^CD49f^low^). The PDX1 positive cells population was sorted from third specimen. (**B**) Histogram of isotype control for IgG2a (PE), (**C**) Phenotype and qualitative analysis of CD45 (PerCP), TER119 (APC-Cy7) expression, (**D**) Histogram of PDX1 PE expression.

**Figure 7 ijms-21-00991-f007:**
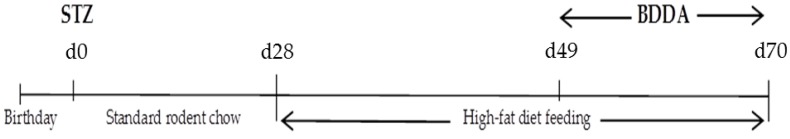
Graphical scheme of the metabolic disorders’ protocols.

**Figure 8 ijms-21-00991-f008:**
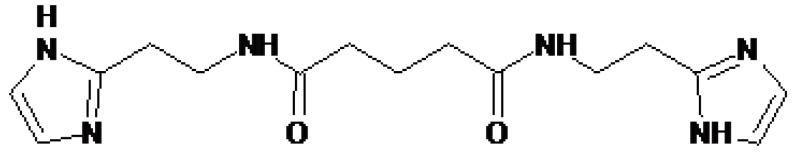
Structural formula of pharmacological compound Bisamide Derivative of Dicarboxylic Acid (BDDA), chemical formular N1,N5-bis[2-(1H-imidazole-2-Il)ethyl]glutaramide.

**Table 1 ijms-21-00991-t001:** Groups of animals in the experiments in vivo.

	Intact Control	Metabolic Disorders	Mice with Metabolic Disorders Treated with BDDA
Males	group 1 (*n* = 20)	group 2 (*n* = 20)	group 3 (*n* = 20)

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
