# Peer review of "Antidiabetic Effects of Bisamide Derivative of Dicarboxylic Acid in Metabolic Disorders"

_ijms, 2020, doi:10.3390/ijms21030991_

Round 1

Reviewer 1 Report

This submission has improved according to comments. However, it needs to conduct the following concerns in addition.

The title seems better to show as “metabolic disorders” than “metabolic syndrome”. Methods must check to show in clear. In conclusion, a decrease in dyslipidemia and obesity mentioned as the main mechanism. Is it suitable? Please check the data shown in this report.

Author Response

Dear Reviewer#1

We thank you for this helpful hint. We now rephrased the title to show as 'metabolic disorders' than 'metabolic syndrome'. 

We now checked and corrected methods.

We now rephrased our conclusions.

Reviewer 2 Report

This is a revised paper, and the authors have fulfilled the comments of the reviewers, and / or responded sufficiently. Therefore, the article should be published.

Author Response

Dear Reviewer#2

We thank you for your valuable time and excellent suggestions and comments.

This manuscript is a resubmission of an earlier submission. The following is a list of the peer review reports and author responses from that submission.

Round 1

Reviewer 1 Report

Bisamide derivative of dicarboxylic acid (BDDA) as target investigated in the present study. I like to give the following comments.

Rationale of the present study is following a previous report in patent(s) as shown in reference 26. Is there any evidence shown the regenerative effect on stem and progenitor cells is benefit for hypogonadism? Model of metabolic disorders needs the reference(s) to support. In general, high-fat diet fed to mice before injection of STZ. Preparation of BDDA in solution must show in clear. Backgrounds for treatment at dose of 10 mg/kg on day 49-70 also need to indicate. Data regarding busulfan remained obscure. Please indicate it in methods before Table 4. In Figure 2, Caro index did not mention in methods. Fasting time is unknown. In Figure 3, TNFalpha did not raise in model. Is it consistent with the previous report? Higher EPO level in this model that needs reference(s) to support. Additionally, GLP-1 levels markedly raised by BDDA. Why? Results in Figure 4G must mention in methods with reference(s). Cell therapy in mice also needs to explain in methods. Mature germ cells in the testicles affected by busulfan must indicate in detail. Condition in mice is so complicated while effectiveness of the treatment explained by a decrease in dyslipidemia and obesity. How to make clear of the potential mechanism(s)? In conclusion, BDDA is effective to improve the engraftment and function of donor stem cells. However, the reason(s) remained unknown.

Author Response

Reviewer 1

Response 1:

We thank the reviewer for their time and valuable comments. We have now revised the manuscript according to the suggestions. Below please find the reviewers’ comments and our responses. The elucidations and the changes have been included into the revised version.

We have corrected the following in the manuscript:

Leydig cells (LCs) play crucial roles in producing testosterone, which is critical in the regulation of male reproduction and development. Low levels of testosterone will lead to male hypogonadism. Two current directions exist for stem cell therapy in male primary hypogonadism. The first method involves differentiating adult Leydig cells from stem cells of various origins from bone marrow, adipose, or embryonic sources. The second method involves isolating, identifying, and transplanting stem Leydig cells into testicular tissue. LC transplantation is a promising alternative therapy for male hypogonadism. However, the source of LCs limits this strategy for clinical applications. Thus far, others have reported that LCs can be derived from stem cells by gene transfection, but the safe and effective induction method has not yet been reported. Here, we report that hypogonadism can treat by effect of BDDA on endogenous stem and progenitor cells.

Streptozotocin (STZ) is a widely used chemical for the induction of experimental diabetes in rodents. STZ is used alone or in combination with other chemicals or dietary manipulations to induce either type 1 or type 2 diabetes, or a mixed type. Models combining the use of STZ and high-fat diets [1], and neonatal STZ injections are used [2]. We modeled metabolic disorder by a single subcutaneous injection of STZ into the withers at a dose of 200 mg/kg in phosphate buffer. When choosing a model, we used publications [2-4], where they use a dose of STZ 200 mg/kg once and the following fatty diet to induce metabolic syndrome. Streptozotocin selectively accumulates in pancreatic beta cells by means of low affinity for the glucose transporter GLUT2 in the plasma membrane. In newborn animals, beta cells are regenerated from precursors located in the channel, and by transdifferentiating beta cells from duct or acinar cells [4, 5]. M. Kataoka and colleagues have shown that in newborn rats, injection of STZ on the day of birth damages beta cells, which is accompanied by regeneration of beta cells, so that about a week after STZ, the mass of beta cells was about half of the original. In mice, after injection of STZ, the regeneration of damaged beta cells proceeds more slowly [5] and is not fully restored, therefore the addition of a fatty diet aggravates the situation, leading to the development of MD.

As requested, we have now added information about the preparation of BDDA and busulfan.

It is known that mouse spermatogonial stem cells could be successfully transplanted to reside and expand in the testis of mice which have been depleted endogenous germ cells by busulfan [6]. The endogenous spermatogenesis in recipient mice can be suppressed at busulfan. We used busulfan induced model for study of the regenerative effect of BDDA on stem and progenitor cells.

We added the formula for calculating the Caro index in methods.

We described the stem and progenitor cell transplantation in 4.16.

We added in reference the mention in methods sections.

The high EPO level in this model was related to obesity or with glucose or lipid profile as described in papers [7, 8]. Besides serum EPO concentration can be trend upwards over age. We propose that a high endogenous EPO level could positively be associated with the components of the metabolic syndrome.

There are only 65% of patients had elevated TNF-α values as the report in the report [9]. In our opinion, it is depend on the stage of metabolic syndrome.

Streptozotocin produced effect on glucose and insulin homeostasis replicate the toxin-induced abnormality in β cell function. The decreased plasma glucose level as well as increased plasma and pancreatic insulin observed in our study indicate that concomitant treatment enthused insulin secretion via GLP-1 stimulation. This assumption is further supported by the pancreatic histology which showed protection of pancreatic β cells from toxic effect of streptozotocin. Concomitant treated animals showed less destruction of pancreatic β cells.

Metabolic disorders increase the risk of developing type 2 diabetes, but they are not diabetes mellitus. In clinical practice, metabolic syndrome is characterized by the presence of at least three of the following five risk factors: abdominal obesity, elevated triglyceride levels, low density lipoprotein cholesterol (HDL), high blood pressure (hypertension), and decreased sensitivity of peripheral  tissues to insulin. Hypogodanism is one of complications of both diabetes and metabolic disorders. Drug therapy for hypogonadism, targeting endogenous tissue-specific stem and progenitor cells, is still limited. The main goal of our work was to evaluate the effects of Bisamide Derivative of Dicarboxylic Acid (BDDA) and its effect on tissue-specific stem cells (beta-cell precursors, spermatogonial stem cells (SSC), endothelial progenitor cells, and multipotent epithelial progenitor cells) in vitro and in vivo. In our article, we confirmed the development of metabolic syndrome in animals, determining the level of cholesterol and triglycerides, fractions of high-density lipoproteins (HDL), low-density lipoproteins (LDL) and very low-density lipoproteins (VLDL), blood glucose, and insulin resistance. In addition, we determined the levels of IL-1ra, IL-1beta, IL-2, 4, 5, 6, 10, 13, 17, 23, TNF-alpha, TGF-1beta, IFN-gamma, EPO, GIP, GLP-1, type 1 collagen. This is a very large material. We have shown that BDDA improves the engraftment and function of donor stem cells after transplantation and such treatment results in outcomes superior to transplantation alone.

References.

Wu J, Yan LJ. Streptozotocin-induced type 1 diabetes in rodents as a model for studying mitochondrial mechanisms of diabetic β cell glucotoxicity. Diabetes Metab Syndr Obes. 2015 Apr 2;8:181-8. doi: 10.2147/DMSO.S82272. Patil MA, Suryanarayana P, Putcha UK, Srinivas M, Reddy GB. Evaluation of neonatal streptozotocin induced diabetic rat model for the development of cataract. Oxid Med Cell Longev. 2014; 2014:463264. doi: 10.1155/2014/463264. Fujii, M., Shibazaki, Y., Wakamatsu, K., Honda, Y., Kawauchi, Y., Suzuki, K., Arumugam, S., Watanabe, K., Ichida, T., Asakura, H. and Yoneyama, H. 2013. A murine model for non-alcoholic steatohepatitis showing evidence of association between diabetes and hepatocellular carcinoma. Med. Mol. Morphol. 46: 141–152 . Kaya-Dagistanli F, Ozturk M. Neonatal streptozotocin administrated rats // Acta Histochem. 2013 Jul; 115 (6): 577-86. doi: 10.1016 / j.acthis.2012.12.007 Kataoka M, Kawamuro Y, Shiraki N, Miki R, Sakano D, Yoshida T, Yasukawa T, Kume K, Kume S. low-dose streptozotocin treatment. // Biochem Biophys Res Commun. 2013 Jan 18; 430 (3): 1103-8. doi: 10.1016/j.bbrc.201212/1230 Shinohara, T., Kato, M., Takehashi, M., Lee, J., Chuma, S., Nakatsuji, N., Kanatsu-Shinohara, M., Hirabayashi, M. Rats produced by interspecies spermatogonial transplantation in mice and in vitro microinsemination. Proc. Natl. Acad. Sci. U.S.A. 103, 13624–13628. 2006 Grote Beverborg N, Verweij N, Klip IT, van der Wal HH, Voors AA, van Veldhuisen DJ, et al. (2015) Erythropoietin in the General Population: Reference Ranges and Clinical, Biochemical and Genetic Correlates. PLoS ONE 10(4): e0125215. doi:10.1371/journal.pone.0125215 Hamalainen P, Saltevo J, Kautiainen H, Mantyselka P, Vanhala M. Erythropoietin, ferritin, haptoglobin, hemoglobin and transferrin receptor in metabolic syndrome: A case control study. Cardiovasc Diabetol. 2012;11: 116-2840-11-116. Bălăşoiu M, Bălăşoiu AT, Stepan AE, Dinescu SN, Avrămescu CS, Dumitrescu D, Cernea D, Alexandru D. Proatherogenic adipocytokines levels in metabolic syndrome. Rom J Morphol Embryol. 2014;55(1):29-33.

Reviewer 2 Report

This is an extensive and interesting study which should be published provided the authors fulfill the following reccommendations:

Tjhe article is quite long and hard to read. The sequence of chapters is somewhat unusual and should be reordered: 1, Introduction, 2. Material & Methods, 3. Results, 4. Discussion, 5. Conclusions The different applied techniques (Cytology, Histology, In vivo, In vitro tests, etc. should be rearranged according to the significance of the findings. I confess, that the material is large and might be hard to arrange. It might be of advantage, that the main issues (findings, manifestation & potneital treatment) would serve a headlines. In order to demonstrate and further investigate in morphology, the authors should briefly mention and cite quantitative tissue measurements such as reported by: KAYSER, Klaus; BORKENFELD, Stephan; KAYSER, Gian. Digital Image Content and Context Information in Tissue-based Diagnosis. Diagnostic Pathology, [S.l.], v. 4, n. 1, dec. 2018. ISSN 2364-4893. ; SILLEM, Martin et al. Phosphohistone 3 (PHH3) and lactate dehydrogenase 5 (LDH5) are expressed in ductal carcinoma in situ of the breast: possible clinical implications. Diagnostic Pathology, [S.l.], v. 3, n. 1, mar. 2017. ISSN 2364-4893. DEROULERS, Christophe et al. Automatic quantification of the microvascular density on whole slide images, applied to paediatric brain tumours. Diagnostic Pathology, [S.l.], v. 2, n. 1, sep. 2016. ISSN 2364-4893.  The quality of the photographs should be improved. Side remarks: The publication of virtual slides would improve the significance of the findings remarkably.

Author Response

Reviewer 2

We thank the reviewer for their valuable time and the excellent suggestions and comment.

Response 2:

Thank you for your review. We are very grateful to the reviewer for your work in reviewing our manuscript. We have revised the manuscript and tried to make the revised version easier to read and follow. Below please find the reviewers’ comments and our responses. The elucidations and the changes have been included into the revised version.

We corrected all the text of our manuscript. We now structured the manuscript as requested. We have also cited the requested papers and implemented this into the manuscript. We have also changed the figures as requested. Additionally, the new reference is included in the paper:

Kayser, K; Borkenfeld, S.; Kayser, G. Digital Image Content and Context Information in Tissue-based Diagnosis. Diagnostic Pathology, [s.l.], 2018. v. 4, n. 1, dec. ISSN 2364-4893.

We investigated the pancreas and testes morphology by standard methods according to the literature data and as described early by us [2]. The hematoxylin and eosin stain is the standard used for microscopic examination of tissues that have been fixed, processed, embedded, and sectioned. In our investigation we used this method to assess the pancreas and testes architecture and inflammatory cells.

Skurikhin, E.G.; Pakhomova, A.V.; Pershina, O.V.; Ermolaeva, L.A.; Krupin, V.A.; Ermakova, N.N.; Pan, E.S.; Kudryashova, A.I.; Rybalkina, O.Y.; Pavlovskaya, T.B.; Litvyakov, N.V.; Goldberg. V.E.; Dygai, A.M. Regenerative Potential of Spermatogonial Stem Cells, Endothelial Progenitor Cells, and Epithelial Progenitor Cells of C57BL/6 Male Mice with Metabolic Disorders. Bulletin of Experimental Biology and Medicine 2017, 163(2), 239-244. doi: 10.1007/s10517-017-3775-1.

Round 2

Reviewer 1 Report

It has been revised according to comments.